# Experimental Study on Fatigue Performance of M60 High-Strength Bolts with a Huge Diameter under Constant Amplitude Applied in Bolt–Sphere Joints of Grid Structures

**Zichun Zhou, Honggang Lei \*, Bin Qiu, Shujia Zhang and Guoqing Wang**

College of Civil Engineering, Taiyuan University of Technology, Taiyuan 030024, China
* Correspondence: lhgang168@126.com

**Abstract:** The high-strength bolts' fatigue is critical for the bolt–sphere joints of grid structures under the action of suspended cranes. High-strength bolts with a huge diameter are used more commonly when the span of grid structures and the burden of suspended cranes increase. However, few works have explored the fatigue performance of high-strength bolts with a huge diameter in bolt–sphere joints of grid structures. Thus, this paper examines M60 high-strength bolts with a huge diameter used in the bolt–sphere joints of grid structures. To this end, an AMSLER fatigue testing machine performed fatigue tests on 27 specimens under constant amplitude. The stress–fatigue life (S–N) curve was obtained by regression analysis and the corresponding constant-amplitude fatigue design method was established. The test results were compared with those of high-strength bolts in other specifications. The results showed that the M60 high-strength bolt has a higher fatigue strength. Furthermore, scanning electron microscopy (SEM) analyzed the macroscopic and microscopic fatigue fracture of the specimens, and the mechanism of the fatigue failure was examined. Our findings provide important experimental data for revising relevant Chinese and international codes and promote the application of high-strength bolts with a huge diameter used in the bolt–sphere joints of grid structures with suspended cranes. This study could fill the gap of fatigue performance data of high-strength bolts in different specifications for bolt–sphere joint grid structures, and provide a basis for further studies.

**Keywords:** fatigue test; grid structure; bolt–sphere joint; high-strength bolt; stress–fatigue life (S–N) curves

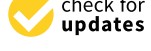



## 1. Introduction

As a kind of leading, large-span, spatial steel structures, grid structures have been universally applied in public and industrial buildings [1]. If the span of a structure is larger than 36 m, a traditional bridge crane cannot be adopted. However, grid structures with suspended cranes enjoy considerable advantages in hangars and industrial buildings because of the flexibility of the arrangement and the adaptability of the process flow; their application prospect is certain to broaden. Figure 1 illustrates the applications of grid structures with suspended cranes.

As the burden, quantity, and operation frequency of suspended cranes increase, the fatigue problem of grid structures intensifies, so it has become a hot topic of research and a complex problem in academic and engineering fields [2]. In most cases, fatigue failure accidents occur suddenly with disastrous consequences. Thus, in-depth and systematic research on fatigue performance can reduce the potential risks and ensure the safety of grid structures.

MERO Company, France, developed bolt–sphere joints in 1942 [1]. In the past 80 years, grid structures using bolt–sphere joints have been widely used globally. Under the load of suspended cranes, the fatigue of bolt–sphere joints primarily occurs at bolted connections. Figure 2 displays a standard bolt–sphere joint.

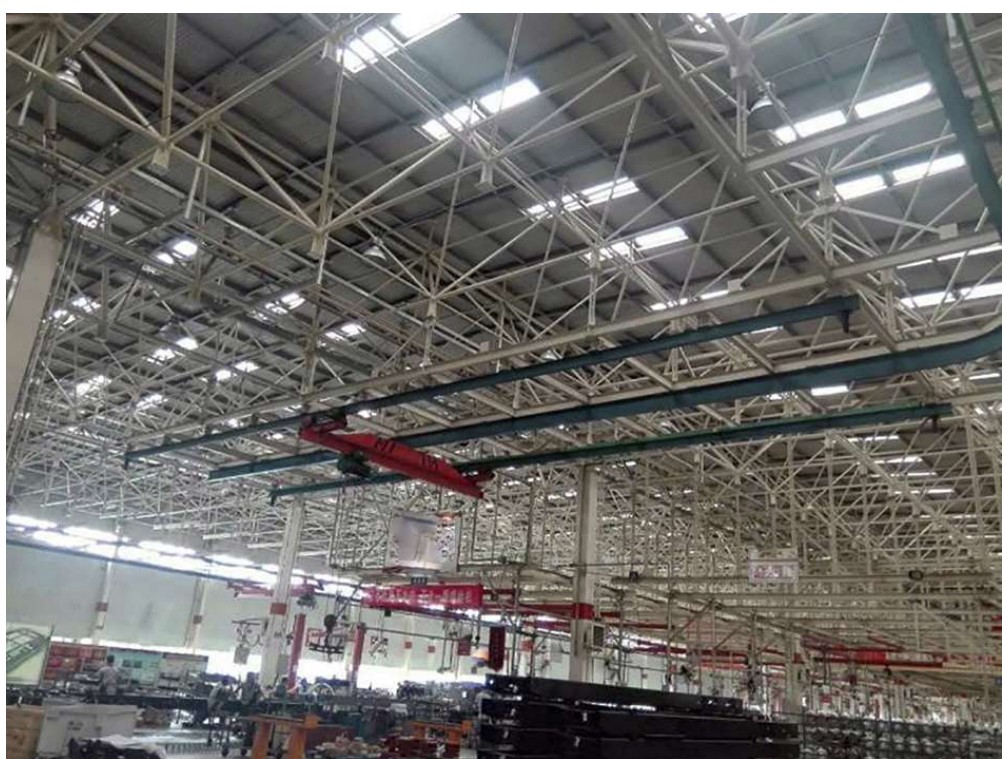

**Figure 1.** The applications of grid structures with suspended cranes.

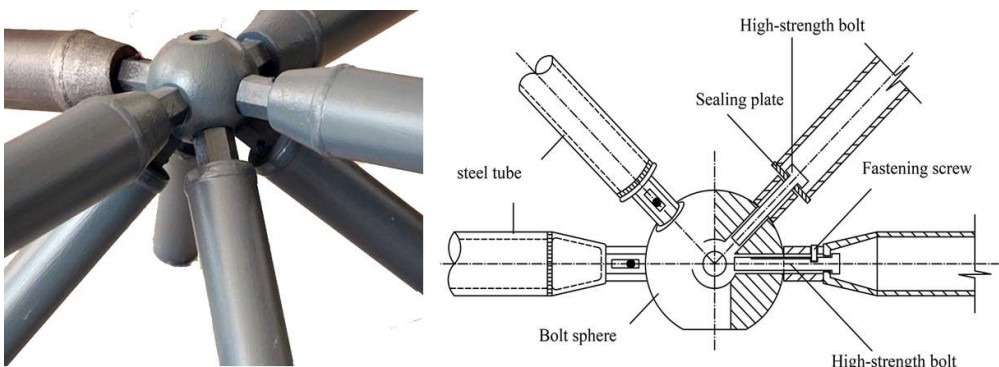

**Figure 2.** A schematic of a bolt–sphere joint.

Nam et al. [3] investigated a new approach to plotting a uniform stress–fatigue life (S–N) curve for three bolts with different dimensions under different external, axial loading conditions and preloading, varying from 44% to 88% of the ultimate tensile stress. Maljaars and Euler [4] drew on many fatigue tests to assess bolts and bolted connections and explored the possibilities of improving current design specifications. Bartsch and Feldmann [5] analyzed fatigue test data and concluded that bolts need to be designed with a diameter as large as possible and placed near the tension flanges possibly close, while end plates must be at least as thick as the diameter of the bolts. Ding B, Zhao Y, Huang Z et al. [6] examined the reliability of the fake tightened-up phenomena of high-strength bolts of bolted spherical joints and developed formulas for estimating the tensile strength of bolted spherical joints with various bolt screwing depths. In another work, Yuan H [7] researched the bearing performance of bolt–sphere joints of bolted spherical joints with random, pitting corrosion damage. Wang et al. [8] found that the appropriate thread rolling process can effectively reduce the tensile residual stress of bolts, and thus improve the fatigue life of bolts. Iordachescu [9] analyzed the failure mechanism of high-strength bolts, and concluded that damage to the zinc coating would reduce the fatigue performance of high-strength



bolts. Yang et al. [10] investigated the relationship between bolt loosening and fatigue failure, and the results could predict the fatigue life in specific circumstances. Bai et al. [11] presented that the fatigue life of some alloys is affected more by material properties than by external loads under uncertain constant amplitude cyclic load.

For research on the fatigue performance of bolt–sphere joints, scholars have conducted some theoretical and experimental studies. For example, Xu and Cui [12] performed 13 uniaxial, tensile fatigue tests on 40 Cr M14 high-strength bolts under constant amplitude and obtained the corresponding S–N curve. Feng et al. [13] considered M24 and M33 high-strength bolts the research objects and conducted constant-amplitude fatigue tests on 12 members of the bolt–sphere-joined grid structure, 12 suspension-point specimens, and 2 bolt–sphere-joined truss specimens. Twenty-one fatigue failure points were determined in the tests, and the related S–N curve was developed. Furthermore, the checking formula was constructed when the stress amplitude was taken as the design parameter. Lei [14] and Yang et al. [15,16] conducted 32 fatigue tests on M20 and M30 high-strength bolts under constant amplitude and collected 35 more test data points generated by Chinese scholars. They determined the S–N curve by unified regression analysis and devised the corresponding constant-amplitude fatigue design method. The allowable nominal stress amplitude for a fatigue cycle ($N$) of $2 \times 10^6$ was 34.0 MPa. In the meantime, the variable-amplitude fatigue tests were conducted on 21 specimens under five loading patterns, and the Corten–Dolan theory estimated the fatigue life under variable amplitude. Yang [17] and Qiu et al. [18] performed 71 constant-amplitude and 25 variable-amplitude fatigue tests on M30 and M39 high-strength bolts with an MTS fatigue testing machine. They determined the related S–N curves and devised the corresponding constant-amplitude fatigue design method. As a result, based on a fatigue cycle of $2 \times 10^6$, the allowable nominal stress amplitude of M30 and M39 bolts was 58.91 and 40.82 MPa, respectively.

In summary, current studies have focused on the high-strength bolts below M39 used for bolt–sphere joints, and the study on the fatigue performance of high-strength bolts with a huge diameter over M39 is still lacking. However, the M60 high-strength bolt has a much larger diameter than bolts of M39 and below. Due to the influences of size effect and manufacture technique, M60 high-strength bolts always feature a discrepancy in fatigue performance. Furthermore, there is also no relevant stipulate in any specification, which restricts the application of large-diameter high-strength bolts in industrial plants. Therefore, it is necessary to study the fatigue performance of large-diameter high-strength bolts. In addition, there is a lack of fatigue design methods in relevant Chinese and foreign codes such as [19]. Therefore, this paper conducted experimental studies on the fatigue performance of M60 high-strength bolts with a huge diameter under constant amplitude and expected to establish their fatigue design method so as to enhance the vigorous development of bolt–sphere joints of suspended crane-equipped grid structures and used in industrial construction.

Eurocode 3 [20] did not validate the S–N curve for high-strength bolts larger than 36 mm in diameter, but this test could fill this gap. It was not including the connection types of bolt–sphere joint of the grid structure in Members and Connections for Fatigue Calculation of Standard for Design' of Steel Structures (GB50017-2017) [21]. The above reasons restrict the development of bolt–sphere grid structure.

This test filled in the relevant content of the above specifications, accumulated the fatigue performance data of high-strength bolts in bolt–sphere joint grid, which was helpful to the prediction of fatigue life and provided a basis for structure designers. The results of this test could promote the development of bolt–sphere joints, especially in large plants with suspension cranes.

## 2. Experimental Procedures

### 2.1. Specimen Design

#### 2.1.1. Bolt Sphere

According to "Bolted Spherical Node of Space Grid Structures" [22], the specification of the adopted bolt sphere was BS300. Its outer diameter was 300 mm, and it was made of steel grade 45. The bolt sphere was forged from a blank steel ball, and it used regular threads. In addition, three couples of screw holes (M60) were machined on the bolt sphere for the axial tensile fatigue test. The structure with three couples of screw holes could improve the utilization rate of the bolt sphere for the tests because each couple could be used to carry out one test failure. Figure 3 depicts the image and specifications of the prepared bolt sphere.

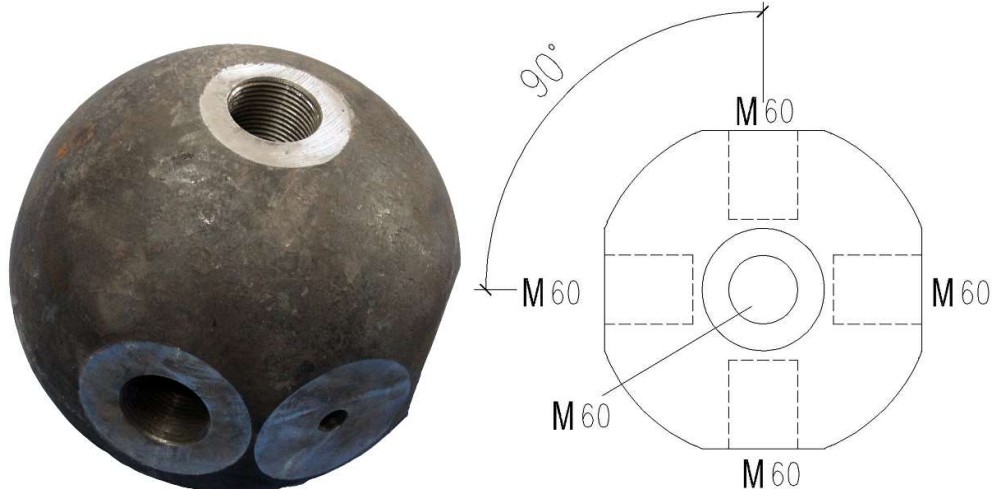

**Figure 3.** The image and specifications of the prepared bolt sphere.

#### 2.1.2. High-Strength Bolt

According to "High-Strength Bolts for Joints of Space Grid Structures", this work used M60 high-strength bolts with a nominal length of 196 mm, a performance grade of 9.8 S, and regular threads; they were made of 40 Cr steel. Figure 4 displays the profile of the M60 bolt used herein. Each bolt was numbered before testing.

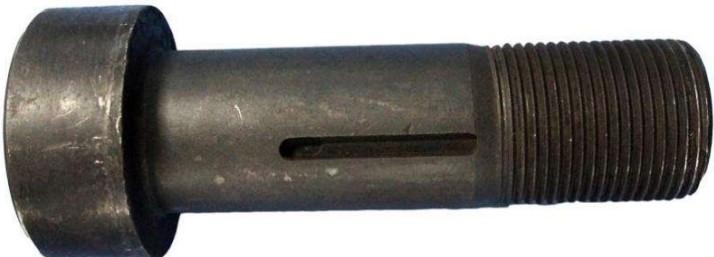

**Figure 4.** The M60 high-strength bolt used in this work.

### 2.2. Material Properties

Three M60 high-strength bolts were randomly selected as the testing specimens and machined according to the relevant regulations described in "Metallic Materials—Tensile Testing—Part 1: Method of Test at Room Temperature" [22]. Figure 5 illustrates the details of the testing specimens. Moreover, the M60 high-strength bolt's chemical composition was analyzed, and an MTS fatigue testing machine performed static tensile tests on the specimens. The bolts fulfill the required standard according to the analysis and test results presented in Tables 1 and 2.

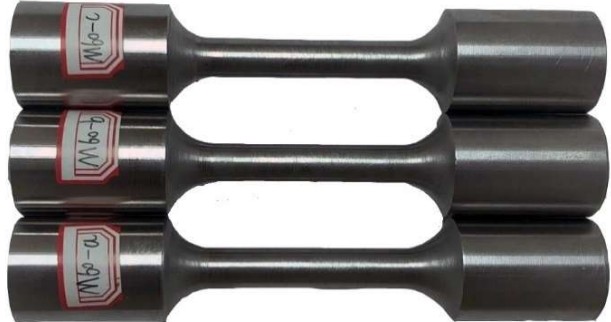

**Figure 5.** The standard specimens of the M60 high-strength bolt for tensile tests.

**Table 1.** The M60 high-strength bolt's chemical composition [23].

| Element | C | Si | Mn | Cr | P | S |
|---|---|---|---|---|---|---|
| Specimen (mass %) | 0.413 | 0.329 | 0.721 | 0.8912 | 0.024 | 0.018 |
| Standard value (mass %) | 0.37–0.44 | 0.17–0.37 | 0.50–0.80 | 0.80–1.10 | ≤ 0.035 | ≤ 0.035 |

**Table 2.** The M60 high-strength bolt's mechanical properties.

| Bolt | $\sigma_{0.2}$ | $\overline{\sigma}_{0.2}$ | $\sigma_b$ | $\overline{\sigma}_b$ | $\delta(\%)$ | $\overline{\delta}(\%)$ |
|---|---|---|---|---|---|---|
| M60-a | 930.6 | | 1047.1 | | 17.6 | |
| M60-b | 922.3 | 931.0 | 1068.6 | 1049.3 | 17.8 | 17.2 |
| M60-c | 929.8 | | 1032.2 | | 16.5 | |

Note: $\sigma_{0.2}$ is the conditional yield strength (MPa); $\overline{\sigma}_{0.2}$ indicates the average conditional yield strength (MPa); $\sigma_b$ represents the ultimate tensile strength (MPa); $\overline{\sigma}_b$ denotes the average ultimate tensile strength (MPa); $\delta$ stands for the elongation rate; $\overline{\delta}$ is the average elongation rate.

## 2.3. Loading Equipment

Grid structures are designed to bear the axial forces, so the axial fatigue test is simple and reasonable, and can reduce the experimental cost. Therefore, a fatigue loading device matching with AMSLER fatigue testing machine was designed, and all specimens' fatigue tests were completed with the designed loading device. In addition, AMSLER testing machine was a Swizz-made hydraulic testing machine with good lateral and axial stiffness. It can automatically record the number of loading cycles during the test and has good control and measure ability while loading required waveform cycles. Figure 6 illustrates the loading equipment used for the AMSLER-1200 fatigue testing machine, Switzerland. The circular frequency of the fatigue testing machine ranged from 0 to 800 rpm with stepless speed regulation.

The loading equipment comprised two box-section steel beams and two force transmission supports, which were able to transfer the test load to the specimen and have a certain ability to correct deviation. Two 50 t hydraulic jacks were equipped symmetrically on two sides of the loading equipment, and the force transmission supports and the bolt–sphere joint were fixed in the middle. With the help of the AMSLER fatigue testing machine, the compressive stress cycles applied to the two hydraulic jacks were transformed into the tensile stress cycles applied to the bolt–sphere joint to achieve the expected fatigue loading. The tests confirmed that the loading equipment was a self-balancing, stable, efficient, and safe system.

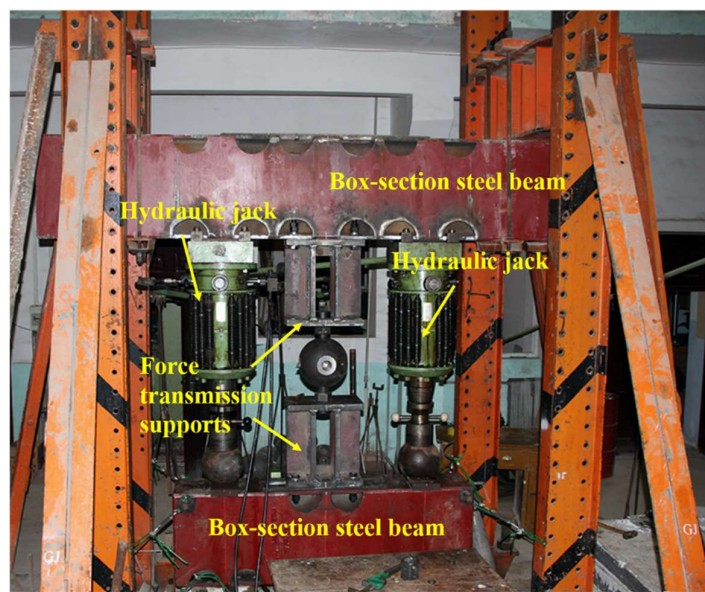

**Figure 6.** The fatigue testing machine and the loading device.

*2.4. Testing Process*

The above loading equipment performed the fatigue tests on M60 high-strength bolts under constant amplitude. The major testing steps are as follows:

(1) Installing and adjusting the loading equipment. Adjusting the position of the specimen to ensure that the upper and lower supports are coaxial with the specimen to satisfy the testing accuracy requirements;

(2) Installing the bolt–sphere joint and ensuring that the M60 high-strength bolt's screwing depth was 1.1 times their nominal diameter, then double check the coaxially;

(3) Applying alternative loads to the specimen using the AMSLER fatigue testing machine according to the predetermined loading grade, closely observing the status of the specimen and the testing apparatus. During the loading process, if any high-strength bolt rod is broken, the loading will be stopped immediately and unloaded carefully. The stress cycles at this time will be recorded, and photos will be taken;

(4) Installing and testing a new bolt when a high-strength bolt fractured; the testing machine automatically stopped upon the bolt fracture. Thus, the fatigue tests under constant amplitude at all predetermined loading levels were complete. A new high-strength bolt specimen would be installed to replace the broken one, the stress amplitude would be reset, and the next round of alternating load would be applied to the new specimen group until fatigue fracture occurred on any high-strength bolt. Then, steps 3 and 4 are repeated.

At present, constant amplitude fatigue test is still the main fatigue research method in the world. Compared with the existing methods, the loading sequence in this test was more reasonable, the load transfer was more accurate, and the test device was self-balanced. This test was scientific, reasonable, and safe. However, due to the influence of test circumstance, the efficiency of the test is relatively low, and the cycle required by the test is relatively long.

## 3. Experimental Results

*3.1. Stress–Fatigue Life Curve*

Generally, the fatigue behavior of a material can be described by the relationship between applied stress level and fatigue life [24–26]. According to "Design of Steel Structures-Part 1–9: Fatigue" [20] and "Standard for Design of Steel Structures" [21], the S–N curve of the bolts could be expressed as:

$$(\Delta\sigma)^m \cdot N = C \tag{1}$$

where $\Delta\sigma$ indicates the stress range, $N$ represents the fatigue life, and coefficients $m$ and $C$ are connected with the bolt type, the stress ratio, and the loading mode. The logarithmic form of the S–N curve can be expressed by:

$$\lg(\Delta\sigma) = A \cdot \lg N + B \tag{2}$$

where $A$ and $B$ are constants determined according to the results of fatigue tests. We calculated the stress range using a 75% confidence level of 95% survival probability for fatigue life, considering the standard deviation. Hence, Equation (3) describes the formula for the S–N curve in both standards:

$$\lg(\Delta\sigma) = A \cdot \lg N + B \pm 2s \tag{3}$$

The fatigue testing of the M60 high-strength bolts under constant amplitude provided 27 data points, among which the fatigue life of bolts M60-29 and M60-30 exceeded 2 million cycles. Table 3 lists the test results. The statistical regression analysis of the data presented in Table 3 determined the M60 high-strength bolt's S–N curves, as delineated in Figures 7 and 8. Equation (4) expresses the formula for the stress–fatigue life curve:

$$\Delta\sigma = 336.9 \cdot N^{-0.413} \tag{4}$$

where the coefficient of determination ($R^2$) of Equation (4) equals 0.9166, indicating a high fitting degree. Equation (5) defines the log–log form of the stress–fatigue life curve with a prediction level of 95%:

$$\lg(\Delta\sigma) = -0.365 \cdot \lg N + 4.099 \pm 0.067 \tag{5}$$

**Table 3.** The fatigue test results of M60 high-strength bolts under constant amplitude.

| Specimen | Stress Ratio (R) | Maximum Stress, $\sigma_{max}$ (MPa) | Minimum Stress, $\sigma_{min}$ (MPa) | Stress Range, $\Delta\sigma$ (MPa) | Fatigue Life, $N$ ($\times 10^4$) |
|---|---|---|---|---|---|
| M60-01 | 0.3 | 315.7 | 98.6 | 217.1 | 8.06 |
| M60-02 | 0.3 | 315.7 | 98.6 | 217.1 | 11.44 |
| M60-03 | 0.3 | 315.7 | 98.6 | 217.1 | 7.42 |
| M60-04 | 0.3 | 315.7 | 98.6 | 217.1 | 8.12 |
| M60-05 | 0.3 | 315.7 | 98.6 | 217.1 | 8.80 |
| M60-06 | 0.3 | 315.7 | 98.6 | 217.1 | 8.47 |
| M60-08 | 0.3 | 161.7 | 51.3 | 110.4 | 44.58 |
| M60-09 | 0.3 | 161.7 | 51.3 | 110.4 | 44.28 |
| M60-10 | 0.3 | 161.7 | 51.3 | 110.4 | 26.40 |
| M60-11 | 0.3 | 161.7 | 51.3 | 110.4 | 54.78 |
| M60-12 | 0.3 | 161.7 | 51.3 | 110.4 | 54.95 |
| M60-13 | 0.3 | 161.7 | 51.3 | 110.4 | 50.29 |
| M60-14 | 0.3 | 161.7 | 51.3 | 110.4 | 33.29 |
| M60-15 | 0.3 | 138.0 | 43.4 | 94.6 | 152.31 |
| M60-16 | 0.3 | 138.0 | 43.4 | 94.6 | 68.66 |
| M60-17 | 0.3 | 138.0 | 43.4 | 94.6 | 62.35 |
| M60-18 | 0.3 | 138.0 | 43.4 | 94.6 | 151.98 |
| M60-19 | 0.3 | 138.0 | 43.4 | 94.6 | 26.01 |
| M60-20 | 0.3 | 138.0 | 43.4 | 94.6 | 34.08 |
| M60-21 | 0.3 | 138.0 | 43.4 | 94.6 | 44.89 |
| M60-23 | 0.3 | 105.5 | 35.5 | 71.0 | 100.79 |
| M60-24 | 0.3 | 105.5 | 35.5 | 71.0 | 124.7 |
| M60-25 | 0.3 | 105.5 | 35.5 | 71.0 | 77.36 |
| M60-26 | 0.3 | 239.2 | 79.7 | 159.5 | 18.94 |
| M60-27 | 0.3 | 239.2 | 79.7 | 159.5 | 15.12 |
| M60-29 | 0.3 | 94.4 | 31.5 | 63.1 | 217.38 |
| M60-30 | 0.3 | 94.4 | 31.5 | 63.1 | 292.51 |

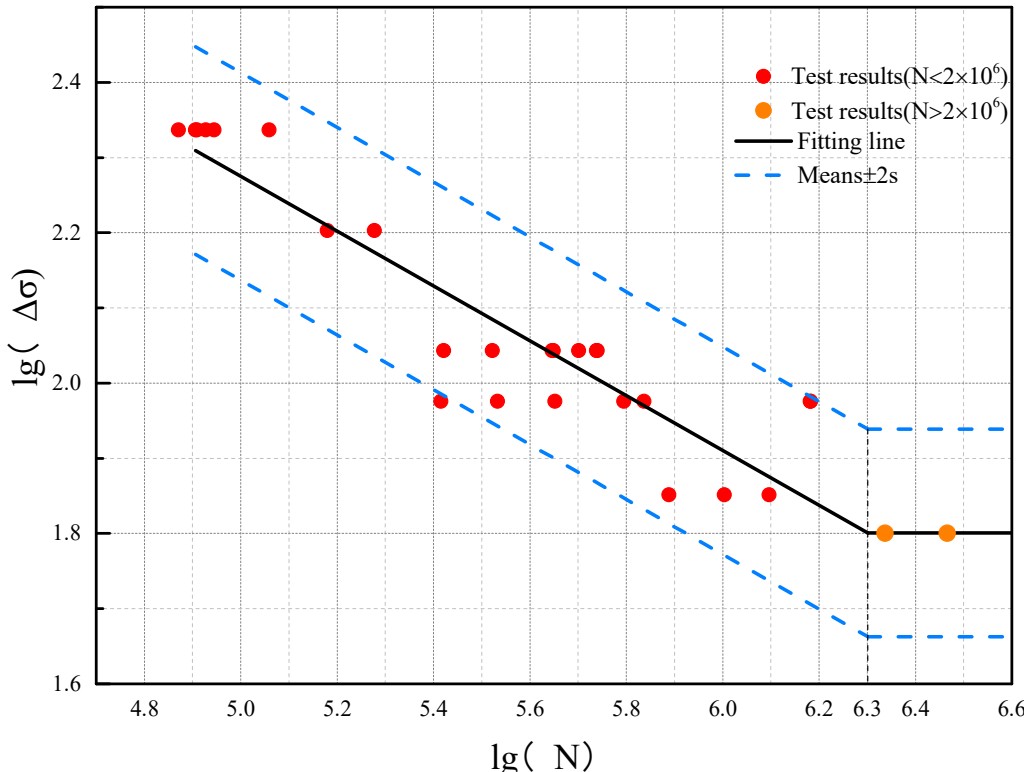

**Figure 7.** The variation of log (Δσ) with log (*N*): The log–log plot of the M60 high-strength bolt's S–N curve.

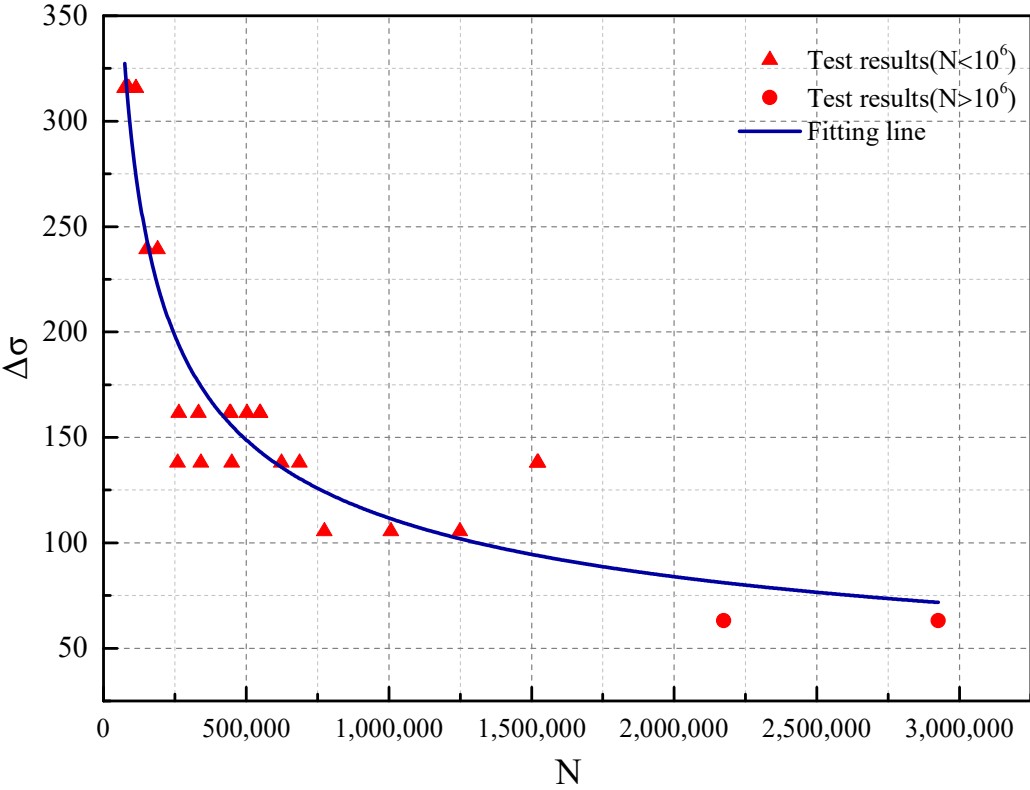

**Figure 8.** The stress–fatigue life relationship: the M60 high-strength bolt's S–N curve.

### 3.2. Failure Mode

Figure 9a illustrates the fatigue fracture of the 27 M60 high-strength bolts. Two failure modes could be observed:

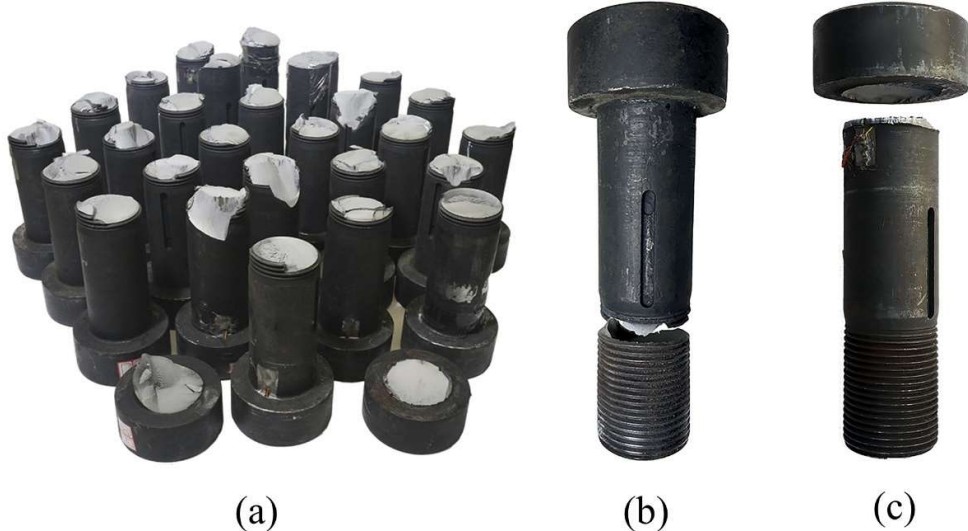

**Figure 9.** The M60 high-strength bolt's failure modes under fatigue loading. (**a**) the fatigue fracture of all specimens; (**b**) The first fatigue failure form of specimens (**c**) The second fatigue failure form of specimens.

Failure mode 1: the fracture occurred on the bolt where its threaded shank engaged with the first thread of the sphere (Figure 9b). A total of 25 specimens failed in this mode. The possible reason for this failure mode was that the primary stress concentration occurred at the fracture position.

Failure mode 2: the fracture occurred at the joint of the shank and head of the bolt (Figure 9c). Only two specimens failed in this mode. This failure mode happened because the secondary stress concentration occurred at the fracture position.

## 4. Discussion

### 4.1. Developing Design Method for Constant-Amplitude Fatigue

Based on the obtained S–N curve, the design method for the fatigue of M60 high-strength bolts under constant amplitude could be developed using the nominal stress amplitude $[\Delta\sigma]$ as the design parameter. The checking formula is given by:

$$\Delta\sigma \leqslant [\Delta\sigma] \tag{6}$$

$$[\Delta\sigma] = \left(\frac{C}{N}\right)^{1/\beta} \tag{7}$$

where $\Delta\sigma$ is the nominal stress amplitude of the high-strength bolt connection: $\Delta\sigma = \sigma_{max} - \sigma_{min}$; $\sigma_{max}$ indicates the maximum stress on the high-strength bolt connection; $\sigma_{min}$ represents the minimum stress on the high-strength bolt connection; $[\Delta\sigma]$ denotes the corresponding allowable stress amplitude; $C$ and $\beta$ are constants: $C = 1.3326 \times 10^6$ and $\beta = 2.4876$; $N$ is the loading cycle number, i.e., fatigue life.

Taking an $N$ of $2 \times 10^6$ as the reference, Equation (7) calculates the allowable stress amplitude of the M60 high-strength bolt as $[\Delta\sigma]_{2\times10^6} = 63.198$ MPa.

Table 4 and Figure 10 compare the experimental results with the literature values for M20 high-strength bolts [11], M30 high-strength bolts, and M39 high-strength bolts [12] to assess the M60 high-strength bolt's fatigue properties. The fatigue strength of M60 high-strength bolts did increase as the diameter of the bolt grew.

**Table 4.** Comparing the fatigue strength of various bolts (MPa).

| Bolt | M60 | M20 | M30 | M39 |
|---|---|---|---|---|
| $[\Delta\sigma]_{2\times10^6}$ | 63.198 | 51.68 | 58.91 | 42.48 |

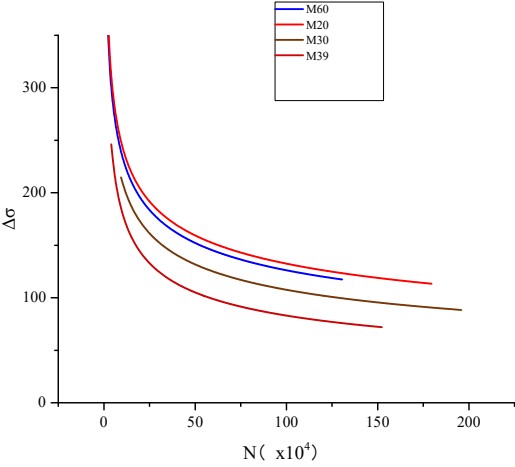

**Figure 10.** Comparing the S–N curve of various bolts.

*4.2. Fatigue Fracture Analysis*

4.2.1. Macroscopic Fracture Morphology Analysis

Figure 11 depicts six macroscopic images of the fatigue fracture, sorted according to the nominal stress amplitude of the high-strength bolt connection. It can be seen that:

- There were single or multiple fatigue sources on the fracture surface;
- The surface was smooth in the crack propagation region (I), and some arc lines could be observed. If the fatigue source was not in a plane, the surface was ladder-shaped; if the fatigue source was basically in the same plane, the surface was relatively flat;
- The surface was rough and fibrillar in the transient fracture region (II);
- As the stress amplitude rose, the area of the crack propagation region decreased gradually, but the area of the transient fracture region increased slowly.

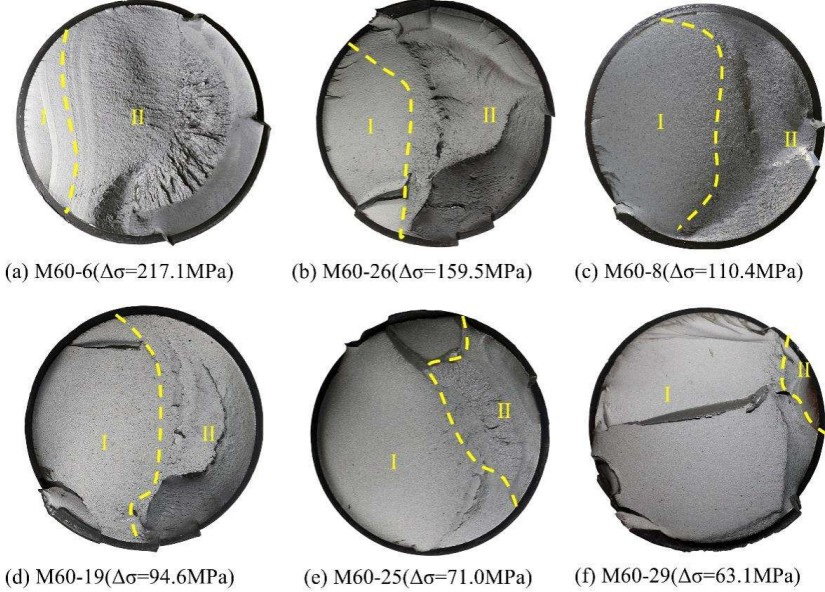

(a) M60-6(Δσ=217.1MPa)  (b) M60-26(Δσ=159.5MPa)  (c) M60-8(Δσ=110.4MPa)

(d) M60-19(Δσ=94.6MPa)  (e) M60-25(Δσ=71.0MPa)  (f) M60-29(Δσ=63.1MPa)

**Figure 11.** Macroscopic fracture morphology of the representative high-strength bolts: I indicates the crack propagation area, and II represents the transient fracture area.

### 4.2.2. Microscopic Fracture Morphology Analysis

The current work analyzed six typical fatigue fractures at different stress amplitudes using scanning electron microscopy (SEM) to further understand the high-strength bolt's fatigue failure mechanism. The first stage of fatigue failure started from the fatigue source, which was always an impurity or a small notch on the surface of the bolt. For example, Figure 12a,c,d shows the fatigue source region starting from the notch on the surface of the bolt. The crack propagated from the notch to the periphery of the bolt. There are apparent inclusions in Figure 12b,e,f, showing radial patterns around the inclusions.

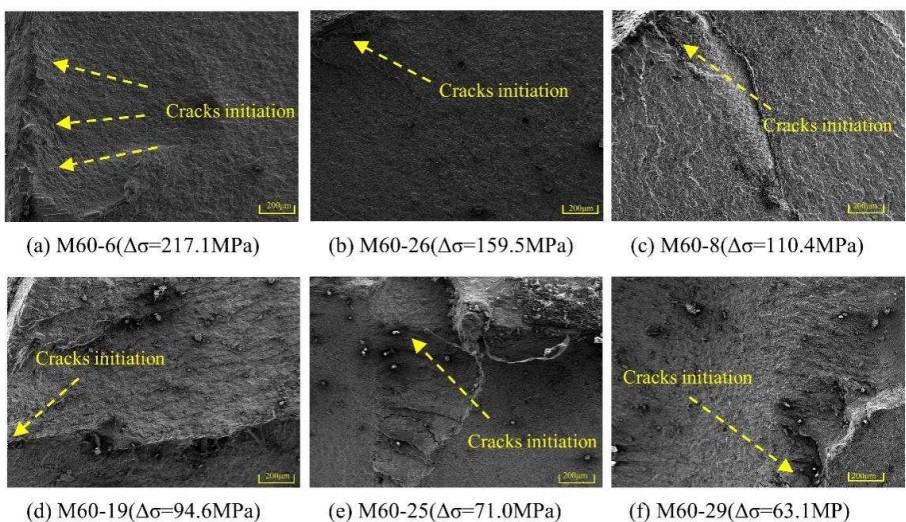

**Figure 12.** The microscopic morphology of the fatigue source at different stress amplitudes.

The second stage of the fatigue failure was the stable propagation of fatigue cracks, and the propagation paths were perpendicular to the direction of the principal stress. Figure 13a,b demonstrates many apparent fatigue striations on the fracture surface, almost parallel. Analyzing the fatigue striations in Figure 13c,d shows that the directions of the striations were not uniform; they were formed by the intersection of cracks propagated from different planes.

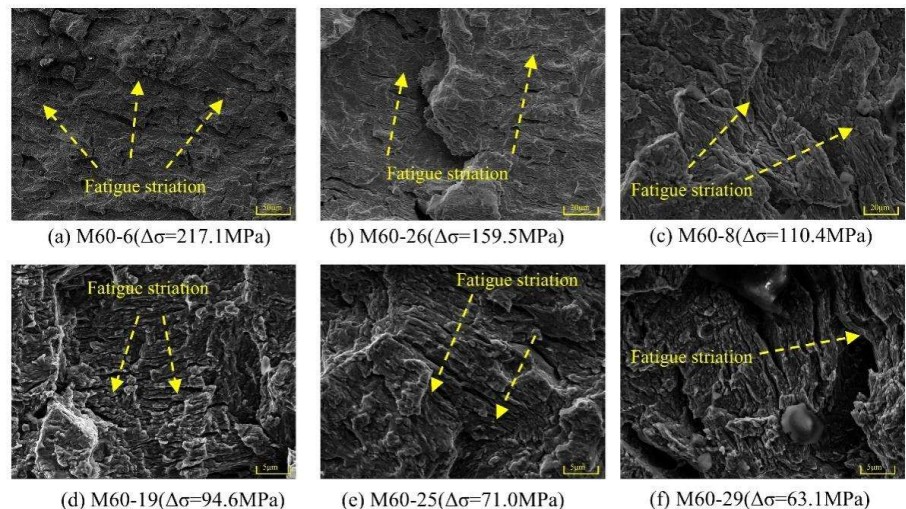

**Figure 13.** The microscopic morphology of the fatigue propagation area under different stress amplitudes.

The third stage of the fatigue failure was the transient fracture. A mixed pattern of fatigue striations and dimples appeared at the junction of the crack instability propagation

area and the fracture area, as presented in Figure 14a,b. Many tear ridges and dimples could be observed in the transient fracture area (see Figure 14c–f), which implied that the material strength was high and the plastic deformation was small, indicating that the final fracture mode of the bolt was a brittle fracture.

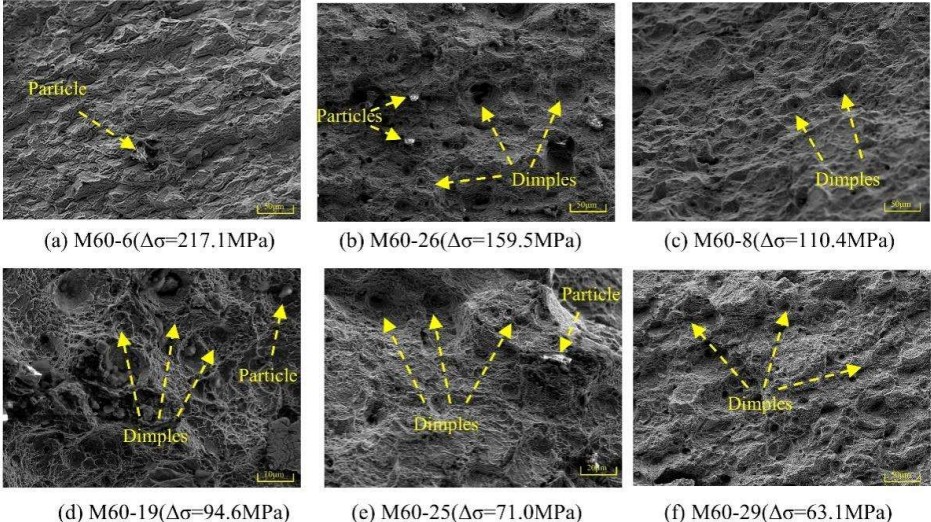

(a) M60-6($\Delta\sigma$=217.1MPa)  (b) M60-26($\Delta\sigma$=159.5MPa)  (c) M60-8($\Delta\sigma$=110.4MPa)

(d) M60-19($\Delta\sigma$=94.6MPa)  (e) M60-25($\Delta\sigma$=71.0MPa)  (f) M60-29($\Delta\sigma$=63.1MPa)

**Figure 14.** The microscopic morphology of the transient fracture region at different stress amplitudes.

The current fatigue test methods were mainly based on constant amplitude fatigue tests, but the fatigue problems encountered in practice were always repeated random loads. This type of load would not cause constant amplitude fatigue stress on components, and almost all fatigue stresses were in variable amplitude periodic. Therefore, it was significant to carry on the variable amplitude fatigue tests. In future works, the fatigue performances of M60 high strength bolts under variable amplitude loads would be further studied.

## 5. Conclusions

This paper used M60 high-strength bolts with a large diameter in the bolt–sphere joints of grid structures and carried out 27 fatigue tests under constant amplitude. The following conclusions can be drawn from the above findings:

1. The M60 high-strength bolt's S–N curve was obtained and expressed in:

$$\lg(\Delta\sigma) = -0.365 \cdot \lg N + 4.099 \pm 0.067$$

2. The design method for the fatigue testing of the M60 high-strength bolts under constant amplitude was developed using the nominal stress amplitude as the design parameter, where $[\Delta\sigma]_{2\times10^6}$ = 63.198 MPa;

3. The fracture morphology analysis revealed the mechanism for the M60 high-strength bolt's fatigue failure. The primary position of the fatigue failure was where the bolt shank engaged with the first thread of the sphere. Furthermore, the fatigue fracture showed typical characteristics of the fatigue source area, the crack propagation area, and the transient fracture area;

4. Compared with M20, M30, and M39 high-strength bolts, the M60 high-strength bolts' fatigue strength did not decline as their diameter increased. This finding could provide an essential scientific basis for the future popularization and application of high-strength bolts with a huge diameter.

**Author Contributions:** Conceptualization, H.L.; methodology, H.L.; software, Z.Z.; validation, Z.Z., S.Z.; formal analysis, Z.Z.; resources, H.L.; data curation, Z.Z.; writing—original draft preparation, Z.Z.; writing—review and editing, Z.Z., S.Z., B.Q. and G.W.; visualization, Z.Z.; supervision, H.L.; funding acquisition, H.L. All authors have read and agreed to the published version of the manuscript.

**Funding:** This research was funded by National Nature Science Foundation of China grant number No. 51578357.

**Conflicts of Interest:** The authors declare that they have no conflict of interest.

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
