# Peer review of "Experimental Study on Fatigue Performance of M60 High-Strength Bolts with a Huge Diameter under Constant Amplitude Applied in Bolt–Sphere Joints of Grid Structures"

_applsci, doi:10.3390/app12178639_

Round 1

Reviewer 1 Report

Please refer attachment

Reviewer 2 Report

Overall, the experiments conducted by the authors are meaningful and well constructed. The calculations, comparisons, and results presented in the paper are interesting to read. While I recommend accepting the paper for publication, there are several areas I would recommend to make the paper better.

1. Another round of proof-reading to make the text more concise and clear and to iron out accidental spaces I saw in several places. 

2. Adding some details regarding M20 vs M30 vs M39 vs M60 for reader not familiar with the field.

3. Adding more details about the testing accuracy requirements mentioned in the testing process section (section 2.4) to illustrate the process in a more rigorous way.

Reviewer 3 Report

-          Line 104. What is the symbol in the brackets?

-          Fig. 7. Is it correct to draw straight line through the last 2 points between 6.3 and 6.5?

-          Does the style of References match with the journal requirements?
